# UniLat3D: Geometry-Appearance Unified Latents for Single-Stage 3D Generation

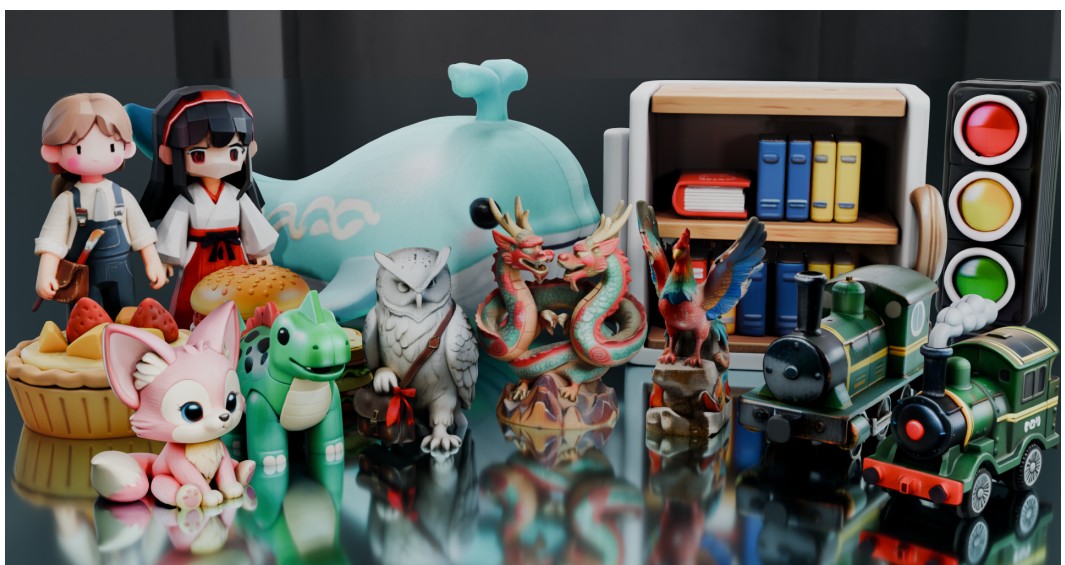

Figure 1: Gallery of UniLat3D. Our method generates high quality 3D assets in seconds.

## ABSTRACT

High-fidelity 3D asset generation is crucial for various industries. While recent 3D pretrained models show strong capability in producing realistic content, most are built upon diffusion models and follow a two-stage pipeline that first generates geometry and then synthesizes appearance. Such a decoupled design tends to produce geometry–texture misalignment and non-negligible cost. In this paper, we propose **UniLat3D**, a unified framework that encodes geometry and appearance in a single latent space, enabling direct single-stage generation. Our key contribution is a geometry–appearance Unified VAE, which compresses high-resolution sparse features into a compact latent representation – **UniLat**. UniLat integrates structural and visual information into a dense low-resolution latent, which can be efficiently decoded into diverse 3D formats, *e.g.*, 3D Gaussians and meshes. Based on this unified representation, we train a single flow-matching model to map Gaussian noise directly into UniLat, eliminating redundant stages. Trained solely on public datasets, UniLat3D produces high-quality 3D assets in seconds from a single image, achieving superior appearance fidelity and geometric quality.

## 1 INTRODUCTION

3D content generation has witnessed rapid growth in recent years, becoming an increasingly essential capability across various applications, including game/film production, virtual/augmented reality, industrial design, and embodied AI. Recent advances in 3D generative frameworks (Zhang et al., 2024c; Hunyuan3D et al., 2025; Lai et al., 2025; Yang et al., 2024c; Zhao et al., 2025; Xiang et al., 2024; Li et al., 2025a; Hong et al., 2023; Zhang et al., 2024b; Ma et al., 2025; Ren et al., 2024a; Zou et al., 2024) have demonstrated impressive progress in synthesizing vivid and realistic

3D assets, while some approaches (Li et al., 2024; Wu et al., 2024b; 2025b; Chen et al., 2025d; Li et al., 2025c; Ye et al., 2025) dive into accurate geometry and fine-grained shape generation.

Despite this rapid progress, the majority of recent high-quality 3D generation frameworks are diffusion-based, and typically adopt a multi-stage design: geometry is generated first, followed by texture or appearance synthesis. This paradigm, rooted in the conventional separation of geometry and appearance, has been adopted by both latent-based pipelines (Xiang et al., 2024) and mesh-based frameworks (Li et al., 2025a; Hunyuan3D et al., 2025), remaining the prevailing design but entailing inherent drawbacks. First, the separate generation introduces an inevitable gap between geometry and appearance, potentially leading to misalignment with the target 3D asset. Second, the two-stage process introduces additional computation budget, *e.g.*, current mesh-based methods (Hunyuan3D et al., 2025) first generate the geometry, and then synthesize the corresponding texture based on both the condition image and geometry generated in the first stage. Notably, the research trajectory in both vision and graphics (Mildenhall et al., 2020; Kerbl et al., 2023) has long favored unification over separation – just as object detection evolved from multi-stage Faster R-CNN (Girshick, 2015) to single-stage YOLO (Redmon et al., 2016). We aim to create a similar unification of geometry and appearance generation, which is expected to offer more convenience and possibilities for exploring 3D generation under a more extensible and unified framework.

To this end, we introduce a unified 3D representation that inherently encodes geometry and appearance in a single latent space, enabling direct single-stage generation. Our key insight is that such a representation is naturally aligned—free from geometry–texture mismatches—and highly efficient, as it avoids redundant intermediate steps. Inspired by TRELLIS (Xiang et al., 2024), we first transform the 3D asset into sparse structured features. A unified variational autoencoder, UniVAE, is designed to compress high-resolution sparse features into a compact latent space, termed **UniLat**. The UniLat can then be efficiently upsampled and sparsified back onto high-resolution latents that serve as a universal basis for decoding into various renderable 3D representations, such as 3D Gaussians (Kerbl et al., 2023) and meshes. Thanks to the simplicity and expres-

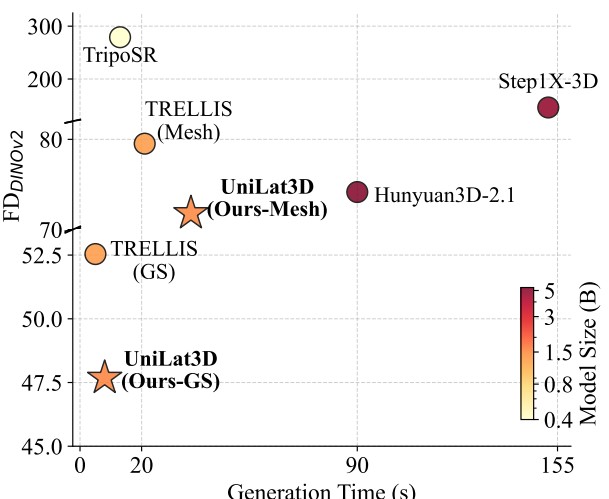

Figure 2: Evaluation on Toys4K (Stojanov et al., 2021). Colors stand for model sizes. Lower generation time and smaller FD$_{\text{DINOv2}}$ indicate better performance, *i.e.* the left bottom corner.

sive design of UniLat, we are able to, for the first time, achieve single-stage 3D generation through one flow-matching model that maps cubic Gaussian noise directly into the geometry-appearance unified latents. Beyond efficiency, UniLat also offers strong extensibility, which can serve as a versatile 3D prior that can be seamlessly integrated into large multimodal models, facilitating cross-modal understanding and generation. Our method, **UniLat3D**, trained only on publicly available datasets, achieves superior appearance fidelity while maintaining strong geometric accuracy, demonstrating the effectiveness of unifying geometry and appearance within a single-stage paradigm.

Our contributions are summarized as follows.

- We propose a novel framework, UniLat3D, which bridges the gap between geometry and appearance by a single diffusion model in high-quality 3D generation.
- A novel UniLat representation is introduced by encoding geometry and appearance into a unified latent space, ensuring high-efficiency feature fusion.
- As in Fig. 2, extensive experiments demonstrate UniLat3D's state-of-the-art performance. We expect our framework to pave a novel way for exploring 3D generation in a more unified and scalable paradigm.

## 2 RELATED WORKS

### 2.1 3D GENERATION BY LIFTING 2D DIFFUSION MODELS

Lifting 2D diffusion models to 3D has been an effective but challenging approach. DreamFusion (Poole et al., 2022) proposes Score Distillation Sampling (SDS) to distill knowledge from the 2D diffusion model into a radiance field. Tang et al. (2023); Yi et al. (2023; 2024); Yin et al. (2023); Ren et al. (2023); Liu et al. (2024); Wang et al. (2023) follow this methodology to generate high-quality 3D Gaussians (Kerbl et al., 2023) in minutes. Meanwhile, Jain et al. (2022); Liu et al. (2023b); Shi et al. (2023); Huang et al. (2024); Long et al. (2023); Liu et al. (2023a); Yang et al. (2024a) fine-tune the image diffusion model to generate multi-view consistent images for synthesizing 3D assets. Video diffusion models (Yang et al., 2024d; Yu et al., 2024; Xing et al., 2024; Ren et al., 2025; Zhao et al., 2024; Gao et al., 2024; Wu et al., 2025a; Liang et al., 2024) are also explored to synthesize high-quality 3D/4D representations (Wu et al., 2024a; Yang et al., 2023; Zhang et al., 2024d; 2025b). However, most of these methods need iterative optimization from different views in each generation process, which takes a non-negligible cost, while hallucination may appear, *e.g.*, Janus phenomenon, due to the lack of 3D priors.

### 2.2 3D GENERATION BY PRETRAINING 3D FOUNDATION MODELS

With the emergence of large-scale 3D datasets, *e.g.*, Objaverse (Deitke et al., 2023), 3D foundation models have been constructed and pretrained to have strong reconstruction and generation abilities.

**3D Foundation Reconstruction Models.** Some feed-forward 3D reconstruction methods (Wang et al., 2024; 2025a; Zhang et al., 2024a; Smart et al., 2024; Li et al., 2025b; Wang et al., 2025b; Yang et al., 2025a), using vision Transformer (Dosovitskiy et al., 2020) (VIT) to encode and match input images' features and recover their relative 3D poses, depths, semantics (Sun et al., 2025; Xu et al., 2025), and other 3D information (Jiang et al., 2025; Smart et al., 2024). Those methods achieve nearly real-time reconstruction given an image sequence, while maintaining accurate pose/depth estimation, and high-quality novel view synthesis.

**3D Foundation Generation Models.** A series of 3D foundation models aims to generate high-quality 3D representations with few or a single image(s) as input in seconds. In the early stage, 3D Generation mainly focuses on structure&shape generation (Ren et al., 2024b; Vahdat et al., 2022) or other latent representation (Yang et al., 2024b). Point-E (Nichol et al., 2022) trains a 3D diffusion model, which is used for generating point clouds from text/image prompts. VecSet (Zhang et al., 2023) proposes to encode 3D assets into vector representations, which are further applied in the geometry diffusion models (Chen et al., 2025d; Hunyuan3D et al., 2025; Lai et al., 2025; Zhang et al., 2024c; Li et al., 2024; Xiong et al., 2025). Then, texture diffusion models (Hunyuan3D et al., 2025; Li et al., 2025a) are followed to color the high-quality mesh. TRELLIS (Xiang et al., 2024) and some recent works (Ye et al., 2025; Wu et al., 2025b; Li et al., 2025c; Chen et al., 2025d) encode multiview images into sparse 3D voxel representations and then decode them into high-quality 3D assets. Several methods are proposed to generate dynamic objects (Chen et al., 2025a; Zhang et al., 2025a; Wu et al., 2025c) or extend 3D generation to the part level (Chen et al., 2025b; Dong et al., 2025; Chen et al., 2025c; Yang et al., 2025b).

We observe that most 3D diffusion models split the generation process into two phases – geometry and appearance. Our research aims to bridge the gap between geometry and appearance in 3D generation by introducing a unified latent space while maintaining the strong performance of 3D diffusion models.

## 3 PRELIMINARY

Recently, TRELLIS (Xiang et al., 2024), a powerful 3D generation framework, has enabled generating high-quality 3D assets in seconds. This is achieved by proposing sparse structured latents (SLATs) $\mathbf{z}_{\text{slat}}$ to represent the 3D asset, which can be decoded into different 3D representations.

**Sparse Structured Latent Representation.** SLAT is defined as a series of latents located at activated surface voxels of the 3D asset, which can be formulated as $\mathbf{z}_{\text{slat}} = \{z_i, p_i\}_{i=1}^{L}$, where $z_i \in R^c$ is a $c$-dimensional latent at the voxel position $p_i \in R^3$, $i = \{1, 2, ...L\}$, $N$ denotes the grid resolution

and $L << N^3$. The coordinates $\{p_i\}$, representing coarse geometry, are computed by voxelizing the 3D asset. The latents $\{z_i\}$, representing appearance and detailed geometry[1], are obtained by aggregating and encoding visual features $\mathbf{f} = \{f_i, p_i\}_{i=1}^L$, extracted by a vision encoder (Oquab et al., 2023) from multiple views of the asset. To learn geometry and appearance respectively, TRELLIS constructs two separate VAE models, *i.e.*, geometry VAE $\{\mathcal{E}_{\text{geo}}, \mathcal{D}_{\text{geo}}\}$ and appearance VAE $\{\mathcal{E}_{\text{app}}, \mathcal{D}_{\text{app}}\}$.

Specifically, the encoder of the geometry VAE transforms activated voxels $\mathbf{p} = \{p_i\}$ to geometry latents $\mathbf{z}_{\text{geo}} \in R^{\frac{N}{s} \times \frac{N}{s} \times \frac{N}{s} \times c}$ with a downsampling factor $s$:

$$\mathbf{z}_{\text{geo}} = \mathcal{E}_{\text{geo}}(\mathbf{p}); \quad \mathbf{p} = \mathcal{D}_{\text{geo}}(\mathbf{z}_{\text{geo}}). \tag{1}$$

The sparse appearance VAEs encodes the sparse 3D features $\mathbf{f}$ into SLATs $\mathbf{z}_{\text{slat}}$, and decodes SLATs into 3D representations $\mathcal{O}$ as:

$$\mathbf{z}_{\text{slat}} = \mathcal{E}_{\text{app}}(\mathbf{f}); \quad \mathcal{O} = \mathcal{D}_{\text{app}}(\mathbf{z}_{\text{slat}}). \tag{2}$$

Note that $\mathcal{E}_{\text{app}}$ only converts $\mathbf{f}$ in the feature dimension. The coordinate information is modeled by $\mathcal{E}_{\text{geo}}$ individually.

**Sparse Structured Latent Generation.** To generate SLAT $\mathbf{z}_{\text{slat}}$, TRELLIS proposes a two-stage generation pipeline. Given the condition image $\mathbf{I}$, TRELLIS builds a geometry generation flow Transformer $\mathcal{F}_{\text{geo}}$ to synthesize geometry latents $\mathbf{z}_{\text{geo}}$ from the noise $\epsilon$. Then, the activated voxels $\mathbf{p}$ can be decoded by $\mathcal{D}_{\text{geo}}$:

$$\mathcal{F}_{\text{geo}} : (\epsilon, t, \mathbf{I}) \rightarrow \mathbf{z}_{\text{geo}}; \quad \mathbf{p} = \mathcal{D}_{\text{geo}}(\mathbf{z}_{\text{geo}}), \tag{3}$$

where $t$ is the denoising timestep. After that, the appearance noise can be added to the activated voxels $\mathbf{p}$ to get the structured noise $\epsilon_{\text{app}} = \{\epsilon_i, p_i\}$. The sparse appearance flow Transformer is optimized to predict $\mathbf{z}_{\text{slat}}$, and the final 3D representation $\mathcal{O}$ can be computed by the appearance decoder $\mathcal{D}_{\text{app}}$:

$$\mathcal{F}_{\text{app}} : (\epsilon_{\text{app}}, t, \mathbf{I}) \rightarrow \mathbf{z}_{\text{slat}}; \quad \mathcal{O} = \mathcal{D}_{\text{app}}(\mathbf{z}_{\text{slat}}). \tag{4}$$

## 4 METHOD

### 4.1 OVERALL FRAMEWORK

**Geometry-Appearance Unified Latent Representation.** Different from TRELLIS (Xiang et al., 2024), which obtains sparse structured latents $\mathbf{z}_{\text{slat}} = \{z_i, p_i\}_{i=1}^L$ in two separate stages, we propose a dense compressed **Lat**ent representation with geometry and appearance **Uni**fied (UniLat) $\mathbf{z}_{\text{uni}} \in R^{M \times M \times M \times d}$ which can be obtained in one single stage, where $d$ is the number of unified latent's channels, $M = \frac{N}{V}$, and $V$ denotes the compression ratio. In the reconstruction stage, we construct a UniLat variational autoencoder (Uni-VAE) $\{\mathcal{E}_{\text{uni}}, \mathcal{D}_{\text{uni},\{\text{gs,mesh}\}}\}$ to encode the 3D assets efficiently. The rich geometry and appearance of an assets $\mathcal{O}$ can be encoded into the UniLat $\mathbf{z}_{\text{uni}}$, which can be further decoded into 3D representations via decoder $\mathcal{D}_{\text{uni}}$ as:

$$\mathbf{z}_{\text{uni}} \leftarrow \mathcal{E}_{\text{uni}}(\mathcal{O}); \quad \mathcal{O} = \mathcal{D}_{\text{uni}}(\mathbf{z}_{\text{uni}}). \tag{5}$$

The unified decoder $\mathcal{D}_{\text{uni}}$ is composed of a upsampling block $\mathcal{D}_{\text{up}}$ and 3D representation decoders $\mathcal{D}_{\text{gs,mesh}}$. For more details, please refer to Sec. 4.2.2.

**Geometry–Appearance Unified Latent Generation.** With geometry and appearance already fused in our UniLat representation $\mathbf{z}_{\text{uni}}$, the generation process becomes naturally streamlined. A unified generative model $\mathcal{F}_{\text{uni}}$ is employed to directly denoise compact noises $\epsilon$ into UniLat $\mathbf{z}_{\text{uni}}$, which can then be decoded by $\mathcal{D}_{\text{uni}}$ into the desired 3D representation:

$$\mathcal{F}_{\text{uni}} : (\epsilon, t, \mathbf{I}) \rightarrow \mathbf{z}_{\text{uni}}; \quad \mathcal{O} = \mathcal{D}_{\text{uni}}(\mathbf{z}_{\text{uni}}). \tag{6}$$

---

[1] Some detailed geometry properties will be decoded from latents $\{z_i\}$, *e.g.* 3D Gaussian positions and mesh vertices. This will be denoted as 'appearance' for short in the following content.

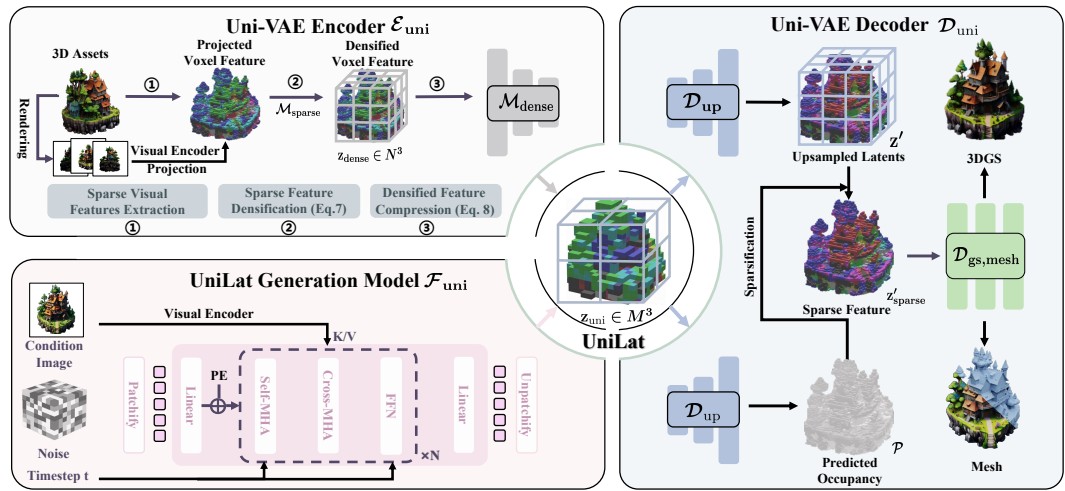

Figure 3: Illustration of the UniLat3D framework. In the reconstruction stage, the encoder of Uni-VAE $\mathcal{E}_{\text{uni}}$ converts the 3D asset $\mathcal{O}$ to the unified latent – UniLat $\mathbf{z}_{\text{uni}}$, which can be directly denoised from noise $\epsilon$ by a single flow model $\mathcal{F}_{\text{uni}}$ in the generation stage. The obtained UniLat can be transformed into target 3D representations by the decoder $\mathcal{D}_{\text{uni}}$.

## 4.2 UniLat Variational Autoencoder

### 4.2.1 Encoder

The encoder of Uni-VAE $\mathcal{E}_{\text{uni}}$ includes two main stages: **Sparse Feature Densification** and **Densified Feature Compression**, and two modules named sparse appearance feature module $\mathcal{M}_{\text{sparse}}$, dense feature compression module $\mathcal{M}_{\text{dense}}$. The key point is the computational budget. We want to get the UniLat $\mathbf{z}_{\text{uni}}$ and begin with the high dimensional voxelized 3D features $\mathbf{f} = \{f_i, p_i\}$. Firstly, we follow TRELLIS (Xiang et al., 2024) to convert 3D object $\mathcal{O}$ to sparse visual features $\mathbf{f} = \{f_i, p_i\}$ and employ sparse appearance feature module $\mathcal{M}_{\text{sparse}}$ to get $\mathbf{z}_{\text{sparse}}$ by $\mathbf{z}_{\text{sparse}} = \mathcal{M}_{\text{sparse}}(\mathbf{f})$. Then, we introduce the **Sparse Feature Densification** process to fill the empty space in the sparse latents and get $\mathbf{z}_{\text{dense}}$. As computation on $\mathbf{z}_{\text{dense}}$ is expensive, we perform **Densified Feature Compression** phase encodes the processed features into lower-resolution compact latents, *i.e.* UniLat $\mathbf{z}_{\text{uni}}$. Finally, the UniLat decoder $\mathcal{D}_{\text{uni}}$ upsamples the compressed unified latents $\mathbf{z}_{\text{uni}}$ back onto high-resolution 3D representations, supporting both 3D Gaussian and mesh outputs.

**Sparse Feature Densification.** Yet we have only obtained the sparse feature $\mathbf{z}_{\text{sparse}}$ with appearance encoded, where the geometry is given by indicating which location is empty. To merge both geometry and appearance information into unified latents $\mathbf{z}_{\text{uni}}$, the structured appearance latents $\mathbf{z}_{\text{sparse}} = \{(z_{\text{sparse,i}}, p_i)\}_{i=1}^{L}$ would be converted to dense features $\mathbf{z}_{\text{dense}}$. All the empty space is assigned with zero features $\{0, p_j\}_{j \neq i}^{N^3 - L}$. Then, the sparse structured uni-latents could be transformed to dense uni-latents:

$$\mathbf{z}_{\text{dense}} : \{\mathbf{z}_{\text{dense}}[p_i] = \mathbf{z}_{\text{sparse,i}}; \mathbf{z}_{\text{dense}}[p_j] = 0\} \tag{7}$$

Here, $z_{\text{dense}}$ is a compact dense latent that includes the whole space information.

**Densified Feature Compression.** Then we use $\mathcal{M}_{\text{dense}}$ to encode both the geometry and appearance features. Similar to 2D/2.5D diffusion models, $\mathcal{M}_{\text{dense}}$ downsample the $\mathbf{z}_{\text{dense}} \in N^3$ to **UniLats** $\mathbf{z}_{\text{uni}} \in M^3$ with downsampling factor $s$:

$$\mathbf{z}_{\text{uni}} = \mathcal{M}_{\text{dense}}(\mathbf{z}_{\text{dense}}). \tag{8}$$

The geometry and appearance features are further fused by the downsampling encoding process, ensuring rich information in the UniLat $\mathbf{z}_{\text{uni}}$ at the low resolution.

#### 4.2.2 DECODER

Uni-Decoder $\mathcal{D}_{\mathrm{uni}}$ includes two modules: upsampling block $\mathcal{D}_{\mathrm{up}}$ and 3D representation decoders $\mathcal{D}_{\mathrm{gs,mesh}}$. The high-resolution dense coordinate and features $\mathbf{z}'_{\mathrm{dense}} \in R^{N^3 \times (C+1)}$ are computed by $\mathcal{D}_{\mathrm{up}}$, then the pruning process is performed on the dense features $\mathbf{z}'_{\mathrm{dense}}$ to obtain sparse features $\mathbf{z}'_{\mathrm{sparse}}$. Finally, representation decoders $\mathcal{O}_{\mathrm{gs,mesh}}$ output the final 3D representations.

**Latent Upsampling and Sparsification.** Given a compact but low-resolution UniLat $\mathbf{z}_{\mathrm{uni}}$, the core challenge is to reconstruct high-quality 3D assets in a detailed manner. To address this, we introduce an upsampling block that lifts $\mathbf{z}_{\mathrm{uni}}$ to higher-resolution latents. Leveraging our geometry–appearance unified representation, we can simultaneously predict voxel occupancy, which guides a pruning step to remove redundant regions among the upsampled latents. This yields a sparse set of high-resolution latents that retain both efficiency and fidelity.

Given UniLat $\mathbf{z}_{\mathrm{uni}} \in R^{M^3 \times d}$, our proposed upsampling blocks $\mathcal{D}_{\mathrm{up}}$ compute the appearance and geometry features at resolution $N$ as :

$$\mathbf{z}'_{\mathrm{dense}} = \mathcal{D}_{\mathrm{up}}(\mathbf{z}_{\mathrm{uni}}). \tag{9}$$

Note that both $\mathbf{z}'_{\mathrm{dense}} = \{\mathcal{P} \in R^{N^3 \times 1}, \mathbf{z}' \in N^3 \times c\}$ are high-resolution dense features. Note that directly performing computation on $\mathbf{z}_{\mathrm{dense}}$ is expensive, so we propose to prune the low-importance area to enhance efficiency. The sparse features $\mathbf{z}_{\mathrm{sparse}}$ are filtered with a signed function:

$$\mathbf{z}'_{\mathrm{sparse}} : \{\mathbf{z}'_i, p_i \mid \mathcal{P}[p_i] > 0\}, \tag{10}$$

**3D Representation Decoders.** Two 3D representation decoders are designed to transform the pruned latents into renderable 3D outputs, *i.e.*, 3D Gaussians and meshes. Both decoders share a backbone of sparse Transformer blocks, similar to TRELLIS, but differ in their task-specific output heads. For 3D Gaussians, the decoder $\mathcal{D}_{\mathrm{gs}}$ maps the latent $\mathbf{z}_{\mathrm{uni}}$ to attributes of 3D Gaussian primitives $\mathcal{O}_{\mathrm{gs}}$ using sparse Transformer blocks and 3D linear projection layers. An additional occupancy head is employed to predicts voxel occupancy, enabling direct supervision of the reconstructed geometry.

For meshes, the decoder $\mathcal{D}_{\mathrm{mesh}}$ progressively upsamples latent features through sparse Transformer and 3D CNN upsampling blocks, followed by a 3D linear output layer that predicts SDF values, voxel-corner deformations, and interpolation weights. The final mesh is extracted with the efficient SparseFlex (He et al., 2025) based on these predicted parameters. To enable multi-scale geometry supervision, occupancy heads are attached at each upsampled resolution. To scale $\mathcal{D}_{\mathrm{mesh}}$ to higher resolutions, we adopt a pruning strategy that removes voxels entirely outside or inside object boundaries, thereby reducing computational overhead. We further introduce a detail augmentation strategy, where depth and normal maps are rendered from zoomed-in camera views with a differentiable rasterizer, enabling the decoder to learn fine-grained surface details from localized partial observations. With these techniques, UniLat3D produces meshes at a resolution of $512^3$, doubling the resolution achieved by TRELLIS ($256^3$).

### 4.3 UNILAT GENERATION MODEL

With the UniLat VAE model, we construct a generation model $\mathcal{F}_{\mathrm{uni}}$ based on rectified flow matching to denoise compact noise $\epsilon$ into condition-followed UniLats $\mathbf{z}_{\mathrm{uni}}$. A single flow Transformer model $\mathcal{F}_{\mathrm{uni}}$ with full attention layers is built to predict the velocity at timestamp $t$ under the noise level as $v = \mathcal{F}_{\mathrm{uni}}(\mathbf{x}_{\mathrm{uni}}, t, I)$ and $\mathbf{x}_{\mathrm{uni}}$ denotes the denoised noise $\epsilon$ and timestamp $t$. The whole flow Transformer optimization process follows the diffusion guidance given condition $\mathbf{I}$ with its condition encoder. The latent features with both geometry and appearance information are denoised. The obtained UniLat $\mathbf{z}_{\mathrm{uni}}$ can be directly fed into the representation decoder $\mathcal{D}_{\mathrm{uni}}$ to predict the final 3D representation $\mathcal{O}$.

### 4.4 OPTIMIZATION

**Uni-VAE.** We use both geometry and appearance supervision to train the $\mathcal{D}_{\mathrm{uni}}$. Following TRELLIS (Xiang et al., 2024), we joint optimize $\mathcal{E}_{\mathrm{uni}}$ and $\mathcal{D}_{\mathrm{gs}}$ with the following loss:

$$\mathcal{L} = \lambda_{\mathrm{l1}}\mathcal{L}_{\mathrm{l1}} + \lambda_{\mathrm{lpips}}\mathcal{L}_{\mathrm{lpips}} + \lambda_{\mathrm{ssim}}\mathcal{L}_{\mathrm{ssim}} + \lambda_{\mathrm{kl}}\mathcal{L}_{\mathrm{kl}} + \lambda_{\mathrm{dice}}\mathcal{L}_{\mathrm{dice}} + \lambda_{\mathrm{reg}}\mathcal{L}_{\mathrm{reg}}. \tag{11}$$

$\mathcal{L}_{l1}$ denotes the L1 color loss, and $\mathcal{L}_{lpips}$ and $\mathcal{L}_{ssim}$ stand for inception-based losses. $\mathcal{L}_{kl}$ is employed for optimizing $\mathcal{E}_{uni}$. $\mathcal{L}_{dice}$ and $\mathcal{L}_{reg}$ are used to supervise geometry and decoded representations.

**Rectified Flow Models.** After trained Uni-VAE, all the UniLat $\mathbf{z}_{uni}$ are predicted by Uni-Encoder $\mathcal{E}_{uni}$. For optimizing the rectified flow Transformer, we mainly follow the CFM Loss. Given encoded latents $\mathbf{x}_{uni}$ and noise $\epsilon$, we minimize the objective function $\mathcal{L}_{CFM}$ (Lipman et al., 2022) as:

$$L_{CFM}(\theta) = \mathbb{E}_{t,x_0,\epsilon}\|v(\mathbf{x}_{uni}, t) - (\epsilon - \mathbf{z}_{uni})\|_2^2. \tag{12}$$

## 5 EXPERIMENTS

### 5.1 IMPLEMENTATION DETAILS

Our framework is implemented in PyTorch (Paszke et al., 2019) and built upon the open-source project TRELLIS (Xiang et al., 2024). FlashAttention-3(Shah et al., 2024) is employed to accelerate Transformer training, yielding a $1.5\times$ speedup. Both VAE and flow models are trained on 64 GPUs within two weeks. More implementation details are provided in the appendix.

### 5.2 EXPERIMENTAL SETUP

**Training Datasets.** UniLat3D is trained exclusively on publicly available datasets. Following the data preparation pipeline of TRELLIS (Xiang et al., 2024), we curate and process approximately 450k high-quality 3D assets from Objaverse (XL) (Deitke et al., 2023), ABO (Collins et al., 2022), 3D-FUTURE (Fu et al., 2021), and HSSD (Khanna* et al., 2023). To enable occupancy supervision at multiple scales, we perform voxelization at each resolution. Additional details on data preprocessing can be found in (Xiang et al., 2024).

**Evaluation Datasets.** The evaluation is performed on two datasets. One is the whole Toys4K (Stojanov et al., 2021) dataset, including 3218 high-quality 3D assets, which is also used in the previous method (Xiang et al., 2024). However, we observe that many samples of Toys4K tend to have simple geometry or appearance details. We construct a more complex dataset for comprehensive evaluation, including 500 high-quality assets collected from the Sketchfab platform and 500 assets sampled from Toys4K. Condition images for qualitative comparisons and user studies are collected from (Chen et al., 2025d; Wu et al., 2025b) or generated via VLMs.

**Evaluation Setups.** For VAE reconstruction evaluation, we use the PSNR, SSIM, and LPIPS metrics. For appearance generation quality, we compute the CLIP (Radford et al., 2021) score – similarity between rendered images and condition images, and FD (Fréchet distance) (Heusel et al., 2017) measured by DINOv2 (Oquab et al., 2023) on 4 views of each generated asset and ground truth images. We evaluate and compare our method with recent SOTA 3D generation models, *i.e*. Hunyuan3D-2.1 (Hunyuan3D et al., 2025), TRELLIS (Xiang et al., 2024), Step1X-3D (Li et al., 2025a), TripoSR (Tochilkin et al., 2024) for image-conditioned generation, and Stable3DGen (Ye et al., 2025) and Direct3D-S2 (Wu et al., 2025b) for geometry generation quality comparison. We report Uni3D (Zhou et al., 2023) and ULIP (Xue et al., 2023) metrics for mesh geometry quality. The Blender rendering pipeline is adopted for all generated mesh assets.

### 5.3 RESULTS

We provide qualitative comparisons in Fig. 4, where our method achieves competitive generation quality and demonstrates stronger alignment with the conditional image, benefiting from the unified representation. Note that Hunyuan3D-2.1 (Hunyuan3D et al., 2025), Step1X-3D (Li et al., 2025a), and TripoSR (Tochilkin et al., 2024) only provide mesh-based results. Importantly, Ours, TripoSR, TRELLIS, and Direct3D-S2 are trained exclusively on publicly available datasets, while other methods leverage additional private data.

Quantitative evaluations on Toys4k (Stojanov et al., 2021) are reported in Table 1. Additional results on our self-collected complex set are provided in the appendix. Compared with other two-stage methods, UniLat3D achieves leading appearance performance, reaching 47.68 in $FD_{DINOv2}$. The CLIP score of 90.87 further demonstrates the effectiveness of UniLat3D in aligning images and 3D assets. In terms of geometry synthesis, our mesh version also achieves competitive results in ULIP (Xue et al., 2023), with a score of 42.69.

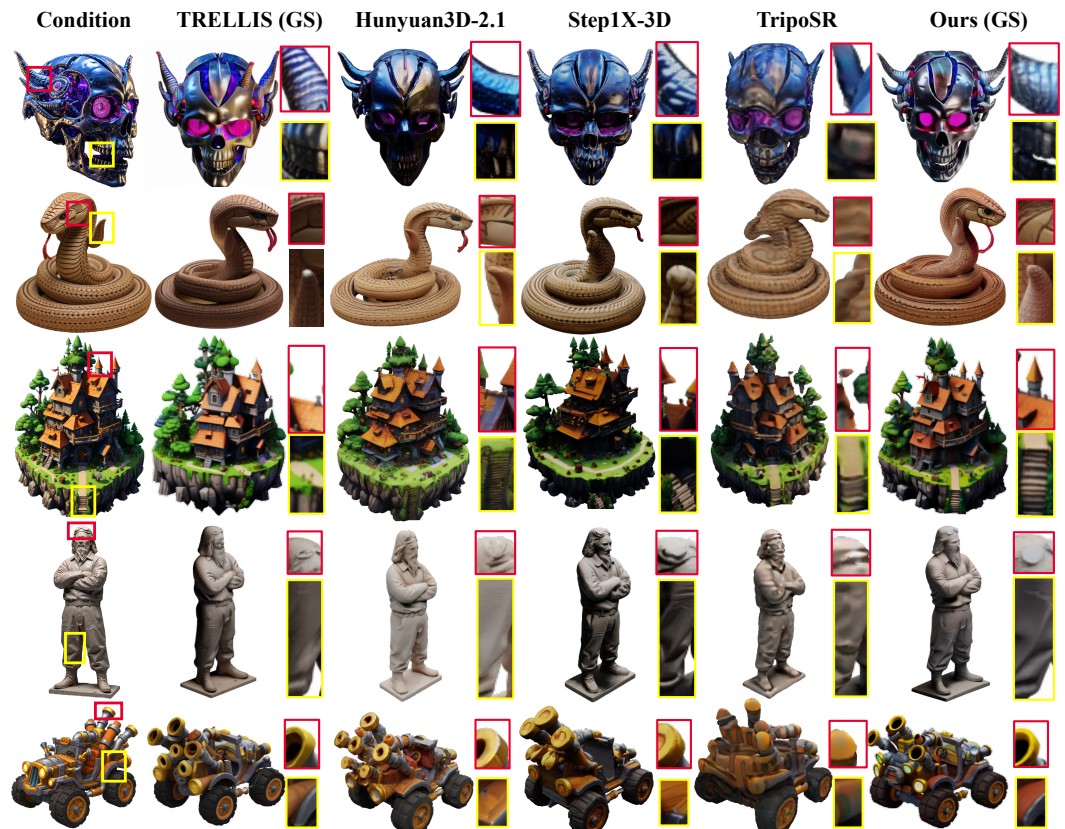

Figure 4: Qualitative comparisons with other methods. Thanks to our unified representation, Uni-Lat3D achieves superior performance and better correspondence with input images.

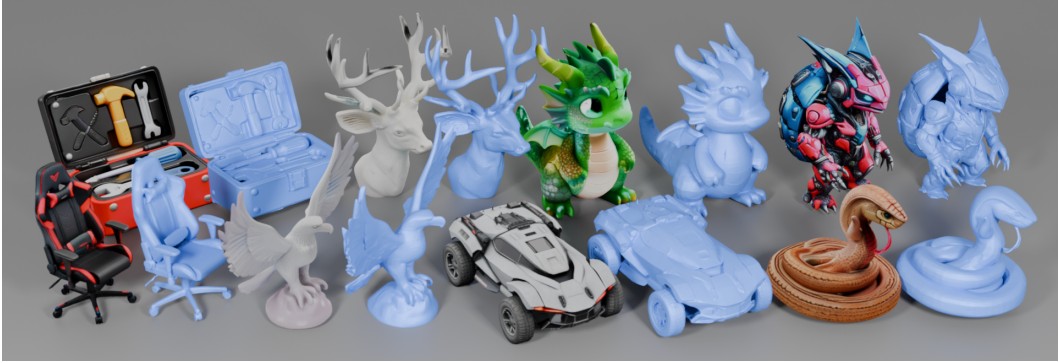

Figure 5: 3D mesh assets generated by our UniLat3D.

Beyond accuracy, UniLat3D also demonstrates notable efficiency: 3D Gaussian generation is completed within 8 seconds on a single A100 GPU and can be further reduced to 3 seconds with FlashAttention-3 (Shah et al., 2024). Mesh generation requires 36 seconds, primarily due to the higher resolution with more vertices and longer post-processing compared to TRELLIS, but remains competitive considering the improved output quality.

Besides, we conducted a user study with 22 participants over 3D assets generated from 23 image prompts. Four models with both geometry and

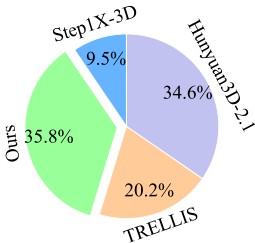

Figure 6: User study on different models.

Table 1: Comparisons on the Toys4K dataset. "Rep." denotes the output representation type.

| Model | Rep. | #Param. | Time | CLIP↑ | FD$_{DINOv2}$↓ | ULIP↑ | Uni3D↑ |
|---|---|---|---|---|---|---|---|
| TripoSR | Mesh | 0.4B | 13s | 88.76 | 279.06 | 35.30 | 30.98 |
| TRELLIS | Mesh | 1.31B | 21s | 87.81 | 79.52 | 42.51 | 37.67 |
| TRELLIS | 3DGS | 1.31B | 5s | 90.70 | 52.54 | – | – |
| Stable3DGen[†][*] | Mesh | 2.63B | 4s | – | – | 40.33 | 35.98 |
| Step1X-3D[†] | Mesh | 4.8B | 152s | 85.85 | 146.08 | 41.37 | 36.51 |
| Direct3D-S2[*] | Mesh | 2.1B | 185s | – | – | 41.51 | 36.64 |
| Hunyuan3D-2.1[†] | Mesh | 5.3B | 90s | 88.44 | 74.16 | 42.67 | **37.74** |
| Ours | Mesh | 1.58B | 36s | 87.93 | 71.81 | **42.69** | 37.62 |
| Ours | 3DGS | 1.55B | 8s | **90.87** | **47.68** | – | – |

[†] Using proprietary or non-public training data.
[*] Only generating geometry without appearance.

appearance generation are involved. For each prompt, participants judged generated assets by both image alignment and object quality, and chose the overall best case. As shown in Fig. 6, UniLat3D received over 35% of the votes, outperforming Huanyuan3D-2.1 and other models.

## 5.4 ABLATION STUDY

**Resolution of Latents.** We explore the latent space of reconstruction quality in Uni-VAE. We train Uni-VAE at different latent resolutions, including $8^3$, $16^3$, and $32^3$. As shown in Table 2, higher UniLat resolutions lead to better reconstruction results. Note that our Uni-VAE achieves similar or even better reconstruction performance than TRELLIS with smaller resolutions. In our experiments, when training the flow Transformer at a higher resolution of 32, the computational cost increases evidently. We would explore more efficient approaches on flow Transformers for higher resolutions in future works, *e.g.*, block-wise computation and lightweight attention.

Table 2: VAE reconstruction results with latents of different resolutions.

| Model | Res. | PSNR↑ | SSIM↑ | LPIPS↓ |
|---|---|---|---|---|
| TRELLIS (Mesh) | $64^3$ | 31.91 | 97.44 | 0.0328 |
| Ours (Mesh) | $16^3$ | 32.35 | 98.03 | 0.0305 |
| TRELLIS (GS) | $64^3$ | 34.74 | 98.52 | 0.0146 |
| Ours (GS) | $8^3$ | 33.51 | 98.13 | 0.0200 |
| Ours (GS) | $16^3$ | 34.80 | 98.49 | 0.0158 |
| Ours (GS) | $32^3$ | **34.92** | **98.53** | **0.0145** |

Another ablation study about the condition visual encoder is included in the appendix.

## 6 DISCUSSION & CONCLUSION

We propose a novel 3D generation framework – UniLat3D to achieve high-quality 3D asset generation in seconds with a single-stage flow model. Apart from that the proposed method unifies geometry and appearance in a single, concise framework, it achieves quite competitive performance compared with popular two-stage methods. We expect our exploration to provide a more convenient and extensible choice to the 3D generation field, *e.g.*, further unifying object and scene generation with the compact unified representation, extending UniLat to 4D representations, and integrating UniLat into large multimodal models *etc*.

However, the UniLat3D model implemented in this paper is still a preliminary exploration. The training data we used just follows TRELLIS, totally from public datasets. Injecting more high-quality data for training will undoubtedly improve the performance and may further scale up the model. Exploring more efficient designs on the flow model would adapt to higher resolutions of latents, leading to more detailed generation results.

## SUBMISSION STATEMENT

### ETHICS STATEMENT

Our UniLat3D only adopts publicly available datasets for training. The generative 3D method may raise potential ethical concerns. Users may use images to create harmful or misleading 3D assets. We encourage the responsible use of our models within legal and ethical boundaries.

### REPRODUCIBILITY STATEMENT

To ensure reproducibility, we introduce implementation details in Sec. A.2 and Sec. A.4.

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

# A APPENDIX

## A.1 USE OF LLMS

In this paper, no LLM models are directly used in the writing of the main text.

## A.2 MORE IMPLEMENTATION DETAILS

**Uni-VAE.** To accelerate and stabilize Uni-VAE training, we initialize $\mathcal{E}_{\text{sparse}}$ and $\mathcal{D}_{\text{sparse}}$ with the pretrained weights from TRELLIS. During the first 240k iterations, only $\mathcal{E}_{\text{dense}}$ and $\mathcal{D}_{\text{up}}$ are optimized, after which the entire Uni-VAE is trained end-to-end for an additional 90k iterations following TRELLIS. For the mesh decoder, we freeze $\mathcal{D}_{\text{uni}}$ and train our high-resolution mesh decoder from scratch. Unless otherwise specified, Adam (Kingma & Ba, 2014) is used with a learning rate of $1 \times 10^{-4}$.

**Mesh Decoder.** We decode the unified latent $\mathbf{z}_{\text{uni}}$ into high-resolution meshes using a high-resolution decoder $\mathcal{D}_{\text{uni,mesh}}$. The decoder first upsamples $\mathbf{z}_{\text{uni}}$ and performs sparsification to obtain $\mathbf{z}_{\text{sparse}}$ (as detailed in Sec. 4.2.2), which is then processed by a stack of sparse Transformer blocks. Two 3D convolutional upsampling layers with skip connections to their corresponding pruning modules further refine $\mathbf{z}_{\text{sparse}}$ to a spatial resolution of $256^3$.

To enable multi-scale geometry supervision and reduce computation, we attach an occupancy head at each upsampled scale $s \in \{128, 256\}$ and supervise it with voxel-level occupancy targets. The occupancy loss is

$$\mathcal{L}_{\text{occ}} = \sum_{s \in \{128, 256\}} \mathcal{L}_{\text{dice}}^{(s)}.$$

The overall training objective for the mesh branch is

$$\mathcal{L}_{\text{mesh}} = \mathcal{L}_{\text{geo}} + \mathcal{L}_{\text{color}} + \mathcal{L}_{\text{reg}} + \mathcal{L}_{\text{occ}}, \tag{13}$$

where $\mathcal{L}_{\text{geo}}$, $\mathcal{L}_{\text{color}}$, and $\mathcal{L}_{\text{reg}}$ follow TRELLIS (Xiang et al., 2024).

Given the computational cost at high resolutions, we adopt a two-stage schedule. *Stage-1* trains $\mathcal{D}_{\text{uni,mesh}}$ up to $256^3$ with the multi-scale occupancy supervision in Eq. equation 13. *Stage-2* adds an independent $256 \rightarrow 512$ upsampling convolutional block with its own pruning head; the newly added block is optimized while keeping the *Stage-1* pathway frozen, using a similar objective as Eq. equation 13.

After training, $\mathcal{D}_{\text{uni,mesh}}$ directly predicts the SparseFlex (He et al., 2025) parameters from $\mathbf{z}_{\text{uni}}$, from which we extract mesh vertices and faces efficiently. Following TRELLIS and prior 3D generation works, we apply lightweight post-processing to remove invisible and degenerate faces and to fill small holes.

**UniLat Flow Transformer.** For training the rectified flow models, we adopt DINOv3 (Siméoni et al., 2025) as the image encoder and apply classifier-free guidance (Ho & Salimans, 2022) with a drop rate of 0.1. The model is first trained for 500k iterations with a batch size of 256 and a learning rate of $1 \times 10^{-4}$, and then fine-tuned for 160k iterations with a batch size of 1024 and a learning rate of $1 \times 10^{-5}$.

## A.3 EVALUATION DETAILS

**Blender Rendering Setups.** For mesh rendering, we mainly use Blender (Blender Foundation, 2025) as a mesh renderer to render high-quality images. We set FOV=40, render resolution=512, and set normalization to each loaded object.

**More Results** We provide evaluation metrics on our self-collected complex test dataset in Tab 3. More comparisons with open-sourced models are shown in Fig. 7. Besides, we provide qualitative comparisons with some commercial models in Fig. 8. Our model still shows notable competitiveness.

Table 3: Comparisons on the self-collected complex test dataset. "Rep." denotes the output representation type, and "#Params" denotes the number of model parameters.

| Model | Rep. | #Param. | Time | CLIP↑ | FD$_{\text{DINOv2}}$↓ | ULIP↑ | Uni3D↑ |
|---|---|---|---|---|---|---|---|
| TripoSR | Mesh | 0.4B | 13s | 88.00 | 369.86 | 33.61 | 30.44 |
| TRELLIS | Mesh | 1.31B | 21s | 86.40 | 164.57 | 41.52 | 37.30 |
| TRELLIS | 3DGS | 1.31B | 5s | 89.67 | 108.27 | - | - |
| Stable3DGen[†][⋆] | Mesh | 2.63B | 4s | - | - | 39.79 | 35.93 |
| Step1X-3D[†] | Mesh | 4.8B | 152s | 84.74 | 210.49 | 40.53 | 36.46 |
| Direct3D-S2[⋆] | Mesh | 2.1B | 185s | - | - | 40.77 | 36.47 |
| Hunyuan3D-2.1[†] | Mesh | 5.3B | 90s | 87.41 | 150.39 | 41.70 | **37.48** |
| Ours | Mesh | 1.58B | 36s | 86.44 | 149.62 | **41.71** | 37.24 |
| Ours | 3DGS | 1.55B | 8s | **89.83** | **97.22** | - | - |

[†] Using proprietary or non-public training data.
[⋆] Only generating geometry without appearance.

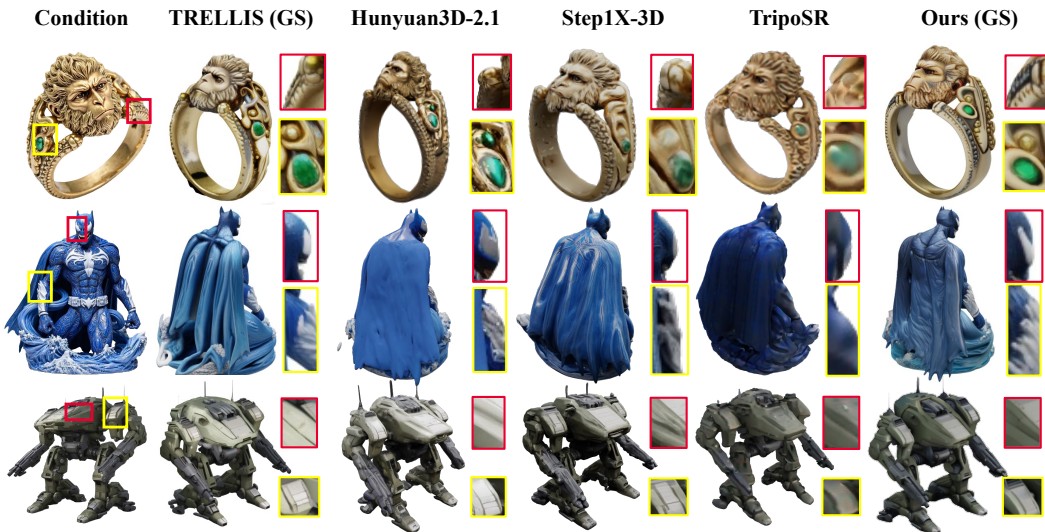

Figure 7: Qualitative comparisons with SOTA open-source models.

## A.4 MODEL ARCHITECTURE

In this section, we mainly provide the model architecture about our Uni-VAE $\{\mathcal{E}_{uni}, \mathcal{D}_{uni}\}$ and UniLat generation model $\mathcal{F}$.

### A.4.1 UNI-VAE

For the sparse encoder $\mathcal{M}_{sparse}$, we mainly follow TRELLIS's configurations to build a sparse Transformer. For the dense encoder $\mathcal{M}_{dense}$, a set of conv3D layers is used as the main architecture. The settings of $\mathcal{E}_{sparse}, \mathcal{D}_{up}$ are shown in Table 4 and details of $\mathcal{E}_{uni}$ are provided in Table 5.

### A.4.2 UNILAT FLOW TRANSFORMER

Structure details about our UniLat flow Transformer $\mathcal{F}_{uni}$ are provided in the Table 6. The main architecture of $\mathcal{F}_{uni}$ is similar to TRELLIS's sparse structure flow Transformer. The input noise $\epsilon$ would be flattened to 1D tensors. Positional encoding is applied to a flattened tensor, and it would be fed to Transformer blocks with self&cross-attention layer and modulated by condition signal & timestamps. Finally, the flattened tensor would be unpatchified to 3D results, the shape is the same as $\epsilon$.

Table 4: Model details of Uni-VAE modules $\mathcal{M}_{dense}, \mathcal{D}_{up}$. "Channels" denotes model channels after each up/downsampled convolution layer.

| Model | ResBlocks | Channels |
|---|---|---|
| $\mathcal{E}_{sparse}$ | 4 | $[32, 128, 512]$ |
| $\mathcal{D}_{up}$ | 4 | $[512, 128, 32]$ |

Table 5: Model details of Uni-VAE modules $\mathcal{M}_{sparse}, \mathcal{D}_{gs,mesh}$.

| Model | Latent Res. | Model Channels | Latent. Channels | Blocks | Attn. Heads | Window Size |
|---|---|---|---|---|---|---|
| $\mathcal{M}_{sparse}, \mathcal{D}_{sparse}$ | 64 | 768 | 8 | 12 | 12 | 8 |

Table 6: Model details of UniLat3D flow Transformer.

| Model | Params | Latent Res. | Latent Channels | Model Channels | Cond. Channels | Blocks | Attn. Heads |
|---|---|---|---|---|---|---|---|
| $\mathcal{F}_{uni}$ | 1.30B | 16 | 32 | 1280 | 1280 | 36 | 32 |

Table 7: Ablation study on the visual encoder for condition images.

| Model | Cond. Encoder | CLIP$\uparrow$ | FD$_{dinov2}\downarrow$ |
|---|---|---|---|
| Ours | DINOV2 | **90.83** | 52.58 |
| Ours | DINOV3 | 90.60 | **49.90** |

## A.5 MORE ABLATION STUDIES

**Visual Encoder of Condition Images**   Recently, DINOv3 (Siméoni et al., 2025) emerges as a strong visual encoder model that could extract high-quality details from the image. We compare the performance between DINOv2 and DINOv3 for encoding condition images. Flow models with different visual encoders are trained for 500 iterations and tested on Toys4K. In our experiments, the flow Transformer with the DINOv3 encoder shows better quality on complex object generation, which leads to a better FD$_{dinov2}$ result as shown in Table 7.

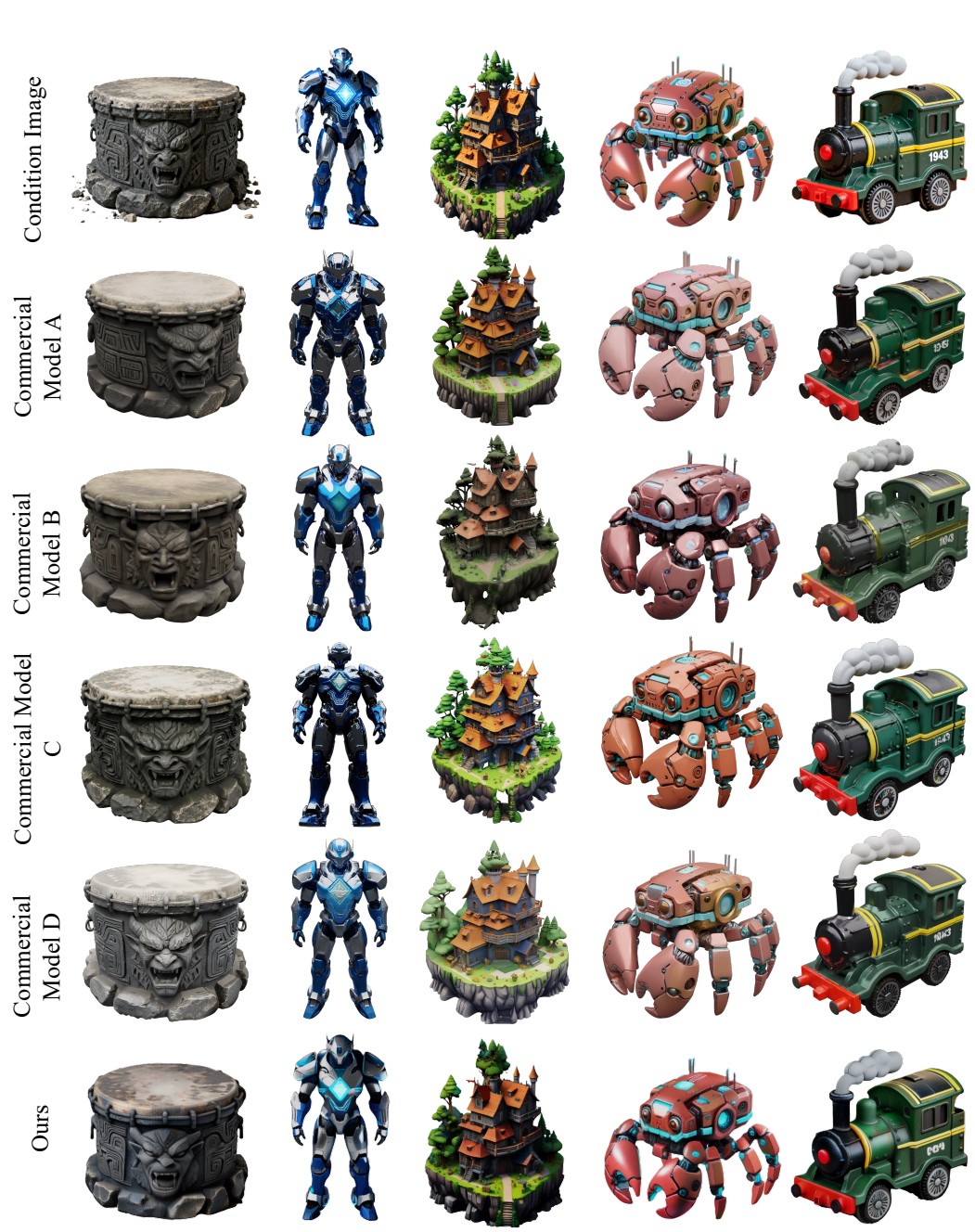

Figure 8: Qualitative comparisons with commercial models. Our UniLat3D shows competitive performance even with only publicly available training data.

