# OpenReview forum: "UniLat3D: Geometry-Appearance Unified Latents for Single-Stage 3D Generation"
_ICLR.cc/2026/Conference — ICLR 2026 Conference Withdrawn Submission_

### Official Review · Reviewer_omD2 · 2025-10-14

**Soundness:** 2
**Presentation:** 3
**Contribution:** 2
**Rating:** 4
**Confidence:** 5

**Summary:**

UNILat3D proposes a unified framework for single-stage 3D generation by encoding both geometry and appearance into a single latent space called UniLat. Unlike previous methods like TRELLIS, which use a two-stage pipeline (first geometry, then appearance), UNILat3D uses a Unified VAE to compress sparse 3D features into a dense, low-resolution latent representation. This allows a single flow-matching model to generate 3D assets directly from noise, without intermediate steps.

**Strengths:**

Uni-VAE: By designing a Unified VAE that encodes both structural and visual information into a single, dense latent space (UniLat), the model is forced to learn the intrinsic correlation between shape and appearance. During generation, the single flow-matching model F_uni makes coherent decisions about what exists where and what it looks like simultaneously. This joint decision-making process ensures that the generated 3D structure is, by design, compatible with its texture.

Efficiency: The unification of representation naturally enables a more efficient generation process, providing a significant practical advantage over more complex, multi-stage counterparts. By fusing geometry and appearance into UniLat, UNILat3D reduces the entire generative process to a single step: mapping noise to the unified latent. This is achieved with one flow-matching model. This architectural simplicity translates directly into speed. The model generates 3D Gaussians in ~8 seconds and meshes in ~36 seconds on an A100 GPU.

The method is trained on public datasets (e.g., Objaverse, ABO) and evaluated on Toys4K and a custom complex dataset. It shows competitive results in both qualitative and quantitative comparisons.

**Weaknesses:**

1. Limited Conceptual Novelty and Heavy Architectural Dependence on TRELLIS

The most significant weakness of UNILat3D lies in its lack of fundamental innovation, as it largely builds upon the existing TRELLIS framework without a substantial conceptual leap.

Architectural Inheritance, Not Revolution: The core pipeline of UNILat3D is nearly identical to that of TRELLIS. Both methods begin by lifting multi-view images into a sparse 3D feature volume. Both rely on sparse Transformers as a core building block for processing these features. The design of the decoders for 3D Gaussians and meshes is also directly inherited. This high degree of architectural overlap positions UNILat3D less as a novel paradigm and more as a significant modification or an extension of the TRELLIS architecture.

Incremental Contribution: The primary proposed innovation is the "UniLat" representation, which is created through a "Sparse Feature Densification" and "Densified Feature Compression" process. While this is a valid technical contribution, it can be perceived as an engineering refinement of the representation rather than a groundbreaking new idea. It essentially converts TRELLIS's explicit, sparse slat into an implicit and dense slat. The core idea of using a unified latent space for 3D generation is appealing, but the implementation here is heavily reliant on repurposing and modifying the components of its predecessor. A truly novel approach might have proposed a more radical architectural departure to achieve unification.

2. Marginal and Ambiguous Performance Improvements

The empirical results fail to demonstrate a decisive advantage over existing methods, particularly its direct baseline, TRELLIS, which undermines the claim that unification is a significantly superior paradigm.

Lack of a impressive factor in quantitative metrics: Examining Table 1 reveals that the performance gains are minimal. For 3D Gaussian generation, UNILat3D achieves a CLIP score of 90.87 versus 90.70. These are modest improvements that do not constitute a clear breakthrough. In mesh-based generation, the ULIP and Uni3D scores are virtually identical to those of TRELLIS and Hunyuan3D-2.1.

**Questions:**

1. The author should clarify the novelty of the paper
2. More experimental results should be included to demonstrate the effectiveness of the proposed method

---

### Official Review · Reviewer_VvUU · 2025-10-16

**Soundness:** 3
**Presentation:** 3
**Contribution:** 2
**Rating:** 2
**Confidence:** 5

**Summary:**

The paper proposed UniLat3D, a single-stage 3D generation framework that aims to unify geometry and appearance generation within a shared latent space, called UniLat. Unlike prior two-stage methods such as TRELLIS that separately model geometry and texture, UniLat3D employs a Unified VAE (Uni-VAE) that first densifies sparse voxel features, compresses them via 3D convolutions, and decodes them into dense latent representations supporting both mesh and 3D Gaussian outputs. A rectified flow model then maps Gaussian noise directly to the unified latents for end-to-end 3D generation. Experiments on Toys4K and other datasets suggest moderate improvements in appearance alignment, though geometric fidelity lags behind leading baselines.

**Strengths:**

* Practical motivation: addresses the inefficiency and misalignment of two-stage 3D pipelines.
* Unified latent representation: enables joint modeling of geometry and appearance, and simplifies pipeline integration.
* Open-data training: uses only public datasets, aiding reproducibility.

**Weaknesses:**

* Limited novelty: the unification mechanism is essentially a dense voxel compression pipeline, and does not offer a clear theoretical or empirical breakthrough. The contribution is more an engineering variant of TRELLIS than a fundamentally new paradigm.
* Dense voxel inefficiency: converting sparse voxels to dense grids defeats the main efficiency advantage of sparse voxel representations and limits scalability to high resolutions (e.g., 1024³+).
* Missing or inconsistent visual comparisons: Fig.4 lacks mesh results of the proposed method, weakening fairness.
* Computational cost: despite claims of efficiency, dense representation and large flow models still demand substantial resources (64 GPUs for 2 weeks).
* Less rigorous baseline selection: TripoSR is a 3D reconstruction model, which is fundamentally mismatched with the task of 3D generation. Furthermore, the comparison omits several native 3D generation pipelines, such as TripoSG [1] and Hi3DGen [2].


[1] TripoSG: High-Fidelity 3D Shape Synthesis using Large-Scale Rectified Flow Models. ArXiv 2025.

[2] Hi3DGen: High-fidelity 3D Geometry Generation from Images via Normal Bridging. ICCV 2025.

**Questions:**

1. The mesh results in Fig. 4 are missing, please include them in the rebuttal to ensure fair visual comparison.
2. How does the latent resolution influence the reconstructed geometry? The current ablation (Tab. 2) only reports the quantitative results on appearance rendering, but lacks metrics like Chamfer Distance or F-score to evaluation the quality of reconstructed geometry.

---

### Official Review · Reviewer_X7Zy · 2025-10-29

**Soundness:** 2
**Presentation:** 3
**Contribution:** 1
**Rating:** 2
**Confidence:** 4

**Summary:**

This paper propose UniLat3D, a framework that generates 3D assets in a single stage by unifying geometry and appearance into a compact latent representation.

**Strengths:**

1. The method achieves good appearance fidelity while using only publicly available training data, unlike competitors that rely on proprietary datasets.

2. The paper is well-written and easy to follow.

**Weaknesses:**

The paper's central claim is that unifying geometry and appearance generation offers significant advantages. However, this claim requires substantial clarification and stronger justification. The paper does not adequately address why unified generation is fundamentally better than the established two-stage paradigm. Separate generation offers several well-documented advantages that are dismissed without proper discussion:

- Independent control over geometry and texture enables flexible editing and iterative refinement
- Compatibility with existing 2D diffusion models and traditional rendering pipelines

The claimed benefits of unification are underwhelming:
- The visual improvements over Hunyuan3D and other baselines are marginal in the provided examples, making it unclear whether unified representation truly solves alignment issues.
- The efficiency gains appear primarily attributable to the smaller model size (1.55B vs. 5.3B) and lower latent resolution (16³ vs. 64³), rather than the unified paradigm itself. Notably, TRELLIS achieves 5s generation for 3DGS compared to the proposed 8s.

The unification itself lacks innovative design. The core contribution reduces to: (1) densifying sparse features via zero-padding (Eq. 7), (2) applying standard 3D convolutions for compression (Eq. 8), and (3) training a diffusion model on the resulting latents. This is a straightforward engineering solution rather than a conceptual breakthrough.

**Questions:**

1. Can you provide a more convincing speed comparison with two-stage models isolating the influence of model size and grid resolution?
2. Demonstrate specific failure cases where two-stage methods produce geometry-texture misalignment that your unified generation resolves.

---

### Official Review · Reviewer_Gsxt · 2025-11-01

**Soundness:** 2
**Presentation:** 2
**Contribution:** 2
**Rating:** 4
**Confidence:** 4

**Summary:**

This paper proposes UniLat3D, a framework for 3D asset generation from a single image. The method is presented as a "single-stage" solution to address the purported geometry-texture misalignment and high costs of two-stage pipelines (which generate geometry then appearance).

The core idea is to modify the VAE from prior work (TRELLIS) to "unify" geometry and appearance into a single compressed latent representation (UniLat). This is done by taking sparse 3D features, densifying them, and then compressing them with a VAE. A single flow-matching model is then trained to generate this UniLat from noise, which is subsequently decoded into 3D Gaussians (GS) or meshes.

The authors claim this unified approach achieves superior performance, particularly in appearance fidelity, while remaining efficient. However, the work appears to be a highly incremental modification of TRELLIS that introduces new bottlenecks and relies on questionable experimental comparisons.

**Strengths:**

1. Addresses a Known Problem: The paper attempts to tackle a recognized limitation (geometry-appearance misalignment) in two-stage 3D generation models.

2. Competitive Metric on GS: The 3DGS variant achieves a strong FD_DINOv2 score (Table 1), suggesting the unified latent may be beneficial for this specific metric, although this comes at the cost of slower inference speed.

**Weaknesses:**

1. The paper's efficiency claims are undermined by its own data. The mesh generation comparison (36s at 512³ vs. 21s at 256³) is invalid due to mismatched resolutions, while the 3DGS comparison (8s vs. 5s) shows a clear regression in speed.

2. The "single-stage" approach introduces a new, severe bottleneck. The authors admit their unified flow model cannot scale beyond a $16^3$ latent resolution without prohibitive computational cost. This directly contradicts the goal of solving the "cost" of two-stage models.

3. The framework is a highly incremental modification of TRELLIS. The core change—the "sparse-to-dense-to-compress" VAE design—is counter-intuitive and presented without any ablation study to justify it over simpler alternatives.

4. In qualitative comparisons (Fig 4, 7), the unified model's results are noticeably smoother and lack the fine-grained texture details visible in competing two-stage models, suggesting the unified latent forces a compromise that sacrifices appearance quality.

**Questions:**

1. Can the authors provide a fair experimental comparison for mesh generation by running their method at the same 256³ resolution as TRELLIS? The current $512^3$ vs $256^3$ comparison is misleading and invalidates any claims about efficiency.

2. The paper admits that a $32^3$ latent space is computationally prohibitive for the flow model. This suggests a severe scalability bottleneck. How can this be considered an "improvement" over a two-stage model, which can scale its geometry and appearance components independently? Doesn't this unified approach make the problem worse?

3. The qualitative results (Fig 4, 7) consistently show a loss of high-frequency texture detail compared to other methods like Hunyuan3D. Is this loss of detail an inevitable consequence of forcing geometry and appearance into a single, compact latent space?

---

### Note · Authors · 2025-11-13

I have read and agree with the venue's withdrawal policy on behalf of myself and my co-authors.